# The Carbon Sink of Mangrove Ecological Restoration between 1988–2020 in Qinglan Bay, Hainan Island, China

**Peihong Jia** [1][ID]**, Weida Huang** [1]**, Zhouyao Zhang** [2,]***, Jiaxuan Cheng** [1] **and Yulong Xiao** [1]

1   School of Public Administration, Hainan University, Haikou 570228, China
2   School of Earth and Environmental Sciences, University of Queensland, Brisbane, QLD 4072, Australia
*   Correspondence: zhouyao.zhang@uqconnect.edu.au

**Abstract:** As the world's largest reactive carbon reservoir, the ocean plays a critical role in global climate change. Among coastal plant ecosystems, mangroves have the highest carbon storage efficiency and are prone to the impact of anthropogenic activities. In this study, taking the mangrove wetland of Qinglan Bay as an example, we extracted information on mangrove coastal surface change from 1988 to 2020 based on multi-temporal Landsat remote sensing data through field ground surveys and laboratory analysis and used the InVEST model to calculate the spatial and temporal structure of blue carbon in the mangrove area to investigate the effects of mangrove change in an ecological restoration context. The result shows that the overall area of mangrove forests exhibited a decreasing trend from 1988 to 2020, and the area of mangroves decreased from 1559.34 ha to 737.37 ha of which 52.71% was transformed into aquiculture, construction, and farm land. Accordingly, the mangrove carbon sinks from 1988 to 2020 were significantly reduced and the carbon stock decreased at an annual tendency from 1,025,901.71 tons to 712,118.69 tons. With the implementation of mangrove restoration, the decline of mangrove forests has decreased since 2003, promoting the stabilization and enhancement of carbon sinks in the mangrove wetlands of Qinglan Bay. The results of this study provide a technical method to evaluate the carbon sink contribution of mangrove wetland restoration in Hainan Province, a scientific basis and methodological innovation to monitor the carbon sink of mangrove cover change in larger regions of China, a theoretical basis to select criteria for mangrove restoration, and a scientific and systematic management strategy for ecological and mangrove restoration on Hainan Island.

**Keywords:** mangrove forest; blue carbon; InVEST model; ecological restoration; Hainan Island

## 1. Introduction

According to the latest data from the United Nations Intergovernmental Panel on Climate Change (IPCC AR6) February 2022, global temperatures have risen by 1.09 °C over the last decades compared to 1850–1900. The Earth's climate is approaching an irreversible turning point [1]. The imbalance of the global carbon cycle has led to the occurrence of extreme climate events [2]. Source reduction is mainly achieved by means of conservation, pollution reduction, and clean energy, while sink enhancement is based on the absorption and elimination of greenhouse gases in the atmosphere [3].

The ocean carbon sink, also known as blue carbon, consists of the carbon absorbed and sequestered by oceanic and coastal zone ecosystems. This carbon is mainly stored in the form of biological biomass and sediment carbon [4]. The oceans are the largest active carbon reservoir on Earth and have an irreplaceable function in regulating climate change.

Mangroves are (i) a unique ecosystem, spanning terrestrial, marine, and wetland ecosystems, (ii) one of the four most productive natural marine ecosystems on earth, and (iii) they have the largest carbon stocks per unit area of any typical ecosystem in the coastal zone [5,6]. Mangroves are also a contiguous area between terrestrial and marine environments and are characterized by high salinity, high temperatures, strong winds and

tides, muddy sediments, and anaerobic soils. In many tropical and subtropical countries, mangrove ecosystems are productive ecosystems that provide economic and environmental benefits to coastal areas. The main roles of mangroves include protecting against storms and tsunamis, regulating water systems, and providing habitat for a variety of fish and other animals. Another important function of mangrove forests is their significant role in climate mitigation activities, with a much stronger carbon sink functionality than terrestrial forests in tropical regions that store and sequester carbon.

Globally, there are approximately 152,000 km$^2$ of mangrove forests, accounting for 0.4% of the terrestrial forest area. The average carbon stock of tropical mangrove wetlands is as high as 1023 Mg Chm$^{-2}$, and the global carbon sink capacity of mangrove wetlands is 0.18–0.228 PgC-a$^{-1}$ [7]. The total area of blue carbon ecosystem habitats in China's coastal zone is 3325.61 km$^2$, with a carbon sequestration rate of 559.72 Gg C-a$^{-1}$ [8]. The mangrove forests of Hainan Island include almost all species of mangrove plants in China, and Hainan Island contains the largest number of mangrove species and preservation areas in China [9]. From the 1950s to the present, the global mangrove forest area was reduced by approximately 30% [10]. The mangrove area in China was reduced by approximately 42.5% from the 1950s to 2001, and the mangrove forest area of Hainan Island decreased by 62% [11]. Large areas of mangrove forests have been destroyed in Sanya, and the remaining mangrove forests are secondary forests or artificial protection forests [12]. From 1959 to 2002, nearly 50% of the natural mangrove forests in Dongzhai Bay were destroyed [13,14]. Historically, the main cause of mangrove destruction has been changes in land use [15]. Studies on the causes of mangrove degradation in China all point to the increased intensity of human activities among which beach polder farming and land reclamation are the main factors that encroach on natural mangrove areas and limit their development [16]. Donato et al. quantified the carbon stocks of an entire ecosystem by measuring the biomass of trees and dead wood, soil carbon content, and soil depth in 25 mangrove sites in the Indo-Pacific region and estimated that mangrove deforestation produces 0.02–0.12 Pg of carbon emissions per year, which is approximately 10% of global deforestation emissions [17]. Miteva et al. conducted the first rigorous large-scale evaluation of the effectiveness of coastal conservation areas, which can reduce mangrove loss by approximately 140 km$^2$ and reduce blue carbon emissions 0.013 GtC·a$^{-1}$ as well [18]. Since 2001, ecological management has gradually gained importance in China, and the implementation of initiatives that promote the return of ponds to forestry and water areas throughout the coastal zone has led to a number of mangrove reconstruction/restoration projects being carried out each year [19]. However, since the 1980s, the overall survival rate of mangrove reconstruction/restoration projects nationwide in China has been below 66%, only reaching approximately 17% in most areas [20]. Due to the lack of systematic and scientific ecological effect monitoring and assessment methods and management systems, artificial mangrove reconstruction/restoration projects have low survival rates and low retention rates and have not achieved the desired restoration effect [21].

The total carbon stock in mangrove wetlands is composed of two parts: one part is stored inside the plants, including the above-ground part of the plant body, the underground roots, and the dead leaves; and the other part is stored in the soil [7]. Ishil and Tateda established a Leaf Area Index (LAI) measurement method based on satellite data and determined the LAI of mangrove plantations in eastern Thailand, as well as the Net Photosynthetic Production and aboveground biomass of eastern Thailand mangrove plantations [22]. Li et al. used Landsat7 TM and radar SAR imagery fusion to conduct the Remote Sensing (RS) estimation of mangrove vegetation in the Zhujiang Estuary [23]. In most research, soil carbon stocks are usually determined using direct measurements, based on field soil profiles sampled to determine the organic carbon content of each layer. Weighting is then used to calculate the organic carbon content of the entire soil profile, and finally the mapped area is used to determine the soil carbon stocks of the whole mangrove wetland [24]. Shi et al. used the Multi-Criteria Evaluation (MCE), Cell Automata (CA), Markov chain, and InVEST model based on the data extracted from the LUCC (Land

Use/Land Cover Change) dataset from 1980 to 2020 to explore the spatial and temporal evolution patterns of terrestrial ecosystem carbon stocks in the Ili Valley from 1980 to 2030 [25]. Based on the practical studies in the abovementioned literature, it can be seen that the InVEST model has the advantages of using less data and less time and can realize the spatial mapping of carbon stock spatial distribution and dynamic changes [26]. The InVEST model is different from the previous monolithic and static service value assessment, and thus it is widely adopted in various regional studies.

The mangrove forest, as an important component of the marine blue carbon sink, is the main ecological carbon pool of Hainan Island. This mangrove forest provides an important ecological support for Hainan Island and functions as a significant economic resource for future carbon trading. Since the middle of the last century, what has happened to the spatial patterns of mangrove cover as a result of anthropogenic activities and the reconstruction/restoration of mangrove forests? What has happened to the carbon cycle of mangrove ecosystems? To address these questions, in this study, the spatial and temporal patterns of mangrove blue carbon distribution on Hainan Island were studied based on the InVEST model, using field surveys and laboratory analyses, and extracting information on mangrove cover changes in the last 30 years based on multi-temporal Landsat RS data so as to investigate the changes in mangrove cover in the context of ecological restoration. The results of the study can provide a basis for the quantitative evaluation of the mangrove carbon sink in Hainan Province. In addition, the findings of this study can provide a technical approach to evaluate the carbon sink contribution of mangrove ecological restoration in Hainan Province and aid future studies on the analysis of mangrove cover change estimations. These results could provide a scientific basis for decision-making and the development of systematic management for mangrove forest restoration projects on Hainan Island.

## 2. Materials and Methods

### 2.1. Study Area

The study area, Qinglan Bay (19°34′ N, 110°45′ E), is located in Wenchang County, northeast of Hainan Island, in the location shown in Figure 1, and it is the second largest mangrove reserve in Hainan Province. The area is dominated by a tropical monsoon climate and consists of a wetland natural reserve that is approximately 3000 ha in size, with a forested area of 2732 ha and 24 species of mangroves, accounting for 85.71% of the 28 species found in China [27].

### 2.2. Data Collection

To reflect the spatial and temporal patterns of mangrove coastal blue carbon distribution on Hainan Island, eight-year representative RS Landsat imageries were selected to extract information on the cover change of mangrove forests in Qinglan Bay, Hainan Island, in the past 30 years. The two main data sources used in this study were LUCC raster charts and carbon density data.

#### 2.2.1. RS Landsat Imagery Data

The RS data used in this study were downloaded from the Geospatial Data Cloud (http://www.gscloud.cn/) (accessed on 1 April 2020). To comprehensively reflect the characteristics of mangrove cover changes in Qinglan Bay, Hainan Island, in the past 30 years, based on the clarity of RS imageries and the availability of information, the data selected included eight-year phases (1988, 1993, 1998, 2003, 2008, 2013, 2018, and 2020) of Landsat imageries, with six TM, two OLI, and a resolution of 30 m. The specific date and imagery details are shown in Table 1.

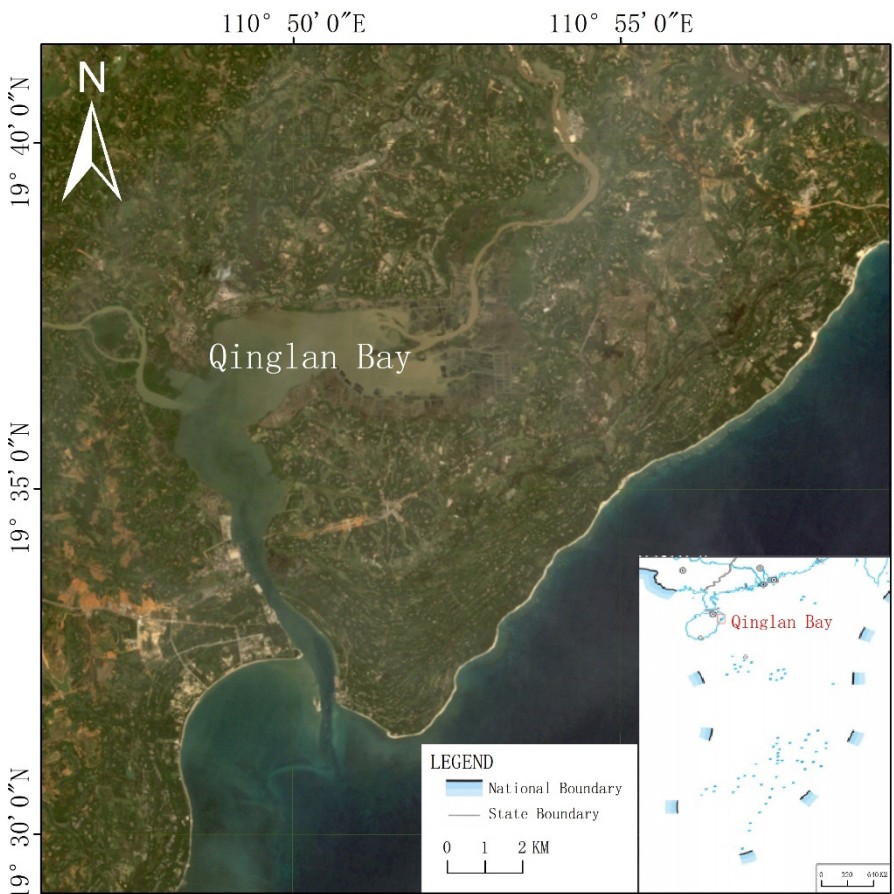

**Figure 1.** Study Area.

**Table 1.** Mangrove RS imageries in Qinglan Bay.

| Sensors | Date | Band | Line No. | Resolution |
|---|---|---|---|---|
| TM | December 1988, April 1993, August 1998, June 2003, December 2008 | 46 | 123 | 30 m |
| OLI | December 2013, April 2018, May 2020 | | | |

### 2.2.2. Carbon Pool Data Extraction

This paper organized the published journal literature on mangrove blue carbon, classified the study areas by latitudes, excluded the carbon pool data with the same or similar latitudes to the distribution of mangrove areas in the study area, and replaced the missing areas with studies conducted in similar latitudes. The final combination of data was extracted from Chen [28] and Xin's [29] studies on China's coastal blue carbon and further obtained the carbon density (pool) table after calculation (Table 2).

**Table 2.** Carbon densities of different land use types (t/hm$^2$).

| Land Types | Above-Ground Organisms | Underground Organisms | Soil | Dead Organic Matters |
|---|---|---|---|---|
| Mangrove | 67.95 | 29.85 | 341.33 | 3.9 |
| Forest | 26.5 | 0 | 30.9 | 0 |
| Aquaculture | 0 | 0 | 12.2 | 0 |
| Constructed land | 4.8 | 0 | 23.3 | 0 |

Mangrove ecosystems consist of four main carbon pools: above-ground and below-ground biomass carbon pools, soil carbon pools, and dead organic matter carbon pools. Due to the difficulty in obtaining required timber data, the total carbon stock calculated in this study took the sum of the four major carbon pools and excluded factors, such as logging volume and the degradation rate of logged products.

### 2.3. Decoding and Extraction of RS Data

Taking the mangrove wetland in Qinglan Bay and its surroundings as the research object, eight periods of satellite RS imageries from 1988 to 2020 were selected, and the ENVI 5.2 software (Boulder, CO, USA) platform was used to decode and extract satellite imageries for eight periods during the last 30 years. The imagery pre-processing procedures included atmospheric radiation correction, imagery cropping, and geometric correction. The pre-processed imageries were basically consistent with the background imageries with a deviation of less than 0.5 pixels, which ensured the accuracy of this study.

### 2.4. The InVEST Carbon Storage and Sequestration Model

The InVEST model is a GIS (Geographic Information System) -based platform developed jointly by Stanford University, the World Wide Fund for Nature, and The Nature Conservancy for ecosystem assessment research [30]. InVEST contains several models for marine, terrestrial, and freshwater analysis among which the Carbon module, which is based on RS data, provides a fast and intuitive method for carbon estimation and is widely used in the calculation of regional carbon sinks [25]. The InVEST model links the carbon storage density to the LUCC raster by inputting the LUCC raster map, logging volume, degradation rate of logging products, and four major carbon sinks (above/below-ground biomass carbon pools, soil carbon pools, and dead organic matter carbon pools) for carbon cycle estimation and value assessment to output the present carbon stocks, the carbon stocks for a specified period of time, and the valuation of economic benefits [30].

### 3. Results

### 3.1. Extraction of RS Imageries and Land Use Data

Combined with the actual survey data, this study used the supervised classification method to extract the RS data of the current land use types in Qinglan Bay, established the land use transfer matrix, and quantitatively analyzed its spatial and temporal dynamics so as to monitor the land use dynamics of mangrove wetlands in Qinglan Bay.

The pre-processed RS imageries were loaded into the ENVI platform and color-composited with R: Band5, G: Band4, and B: Band3. According to research needs, the land in the Qinglan Bay area was classified into five categories: mangrove forest, aquaculture land, forest land, construction land, and water areas. By reviewing local historical archives and topographic maps, combined with Google Earth, different types of land features were selected as supervised classification samples. The compute ROI separability function was used on the ENVI platform, and the Jeffries–Matusita and Transformed Divergence were used to indicate the separability of the selected samples, with a sampling value range of [0, 2.0]. If the value of the sample was greater than 1.9, then it was available for use. Taking the 2008 sample data as an example, the separability values are shown in Table 3.

After selecting a neural-network-based, pattern-recognition-based vector classifier to derive supervised classification results, the sporadically distributed ground classes were organized, or clustered. The Change Detection Statistics tool was used on the ENVI platform to generate a land use type transfer matrix.

### 3.2. Spatial and Temporal Dynamics of LUCC in Qinglan Bay Mangrove Wetland

Eight periods of the satellite RS imageries of the mangrove wetlands in Qinglan Bay, Hainan Province were selected from the past 30 years, and the land use data were extracted using ENVI 5.2. The spatial distribution of land use and dynamic changes in the areas in

the coastal zone of Qinglan Bay from 1988 to 2020 were calculated, as shown in Figure 2 and Table 4. The area of mangroves declined from 1559.34 ha to 737.37 ha in the initial phase of which 52.71% was transformed into other land categories. Most of the mangrove forests were transformed into aquaculture, while some mangroves were converted into constructed land.

**Table 3.** Severability of samples in 2008.

| Samples | Jeffries–Matusita, Transformed Divergence |
|---|---|
| Mangrove | Forest: (1.99964123 1.99999989) Aquaculture: (1.99995880 2.00000000) Water: (2.00000000 2.00000000) Constructed Land: (1.99995791 2.00000000) |
| Forest | Mangrove: (1.99964123 1.99999989) Aquaculture: (2.00000000 2.00000000) Water: (2.00000000 2.00000000) Constructed Land: (1.99999665 2.00000000) |
| Aquaculture | Mangrove: (1.99995880 2.00000000) Forest: (2.00000000 2.00000000) Water: (1.90403571 2.00000000) Constructed Land: (1.97534302 1.99831364) |
| Water | Mangrove: (2.00000000 2.00000000) Forest: (2.00000000 2.00000000) Aquaculture: (1.90403571 2.00000000) Constructed Land: (1.99999579 2.00000000) |
| Construction Land | Mangrove: (1.99995791 2.00000000) Forest: (1.99999665 2.00000000) Aquaculture: (1.97534302 1.99831364) Water: (1.99999579 2.00000000) |

From 1988 to 1993, fenced aquaculture emerged in Xiatian Village, Shatou Port, and Qingtou Village, and the aquaculture area gradually expanded. Because the tideline in the RS imagery in 1993 was lower than that in 1988, some water bodies transitioned to mangrove forests, and the number of mangrove forests also increased slightly.

From 1993 to 1998, the increasing farming activities around Lujue Village, Danchang Village, Qunjian Village, Paigang Village, Wenhu Village, Baobiao Village, Xiayang Village, and Wenjiao River caused a significant disturbance to the mangrove ecosystem, with a rapid decline of mangrove areas and expansion of the aquaculture area.

From 1998 to 2003, no new large-scale waters were added within the mangrove forest of Qinglan Bay. Compared with 1993–1998, the growth rate of the mangrove area slowed, but the growth rate of the farming area remained rapid. Mangroves far from coastal areas were transformed into construction land, and the decline of mangrove area reached its peak. Governments at all levels took various measures to strengthen mangrove protection and vigorously promote mangrove protection and restoration.

From 2003 to 2008, mangroves, water areas, and forests were transformed into expanding aquaculture areas, although this process slowed as mangrove ecological restoration projects were implemented.

From 2008 to 2013, the mangrove ecological restoration projects began to yield prominent results, with only a small increase in aquaculture area, and the decline in the area of mangroves continued to slow.

From 2013 to 2018, the scope of aquaculture waters, mainly in the Wenjiao River basin, gradually decreased, while mangrove forests increased with the implementation of the Regulations on the Protection and Development of the Coastal Zone of Hainan Special Economic Zone. The comprehensive management and restoration of coastal zones began to bring back mangrove lands.

However, from 2018 to 2020, there was a small decline in mangrove forests in Wenjiao River and Qichai River, as well as their adjacent sites.

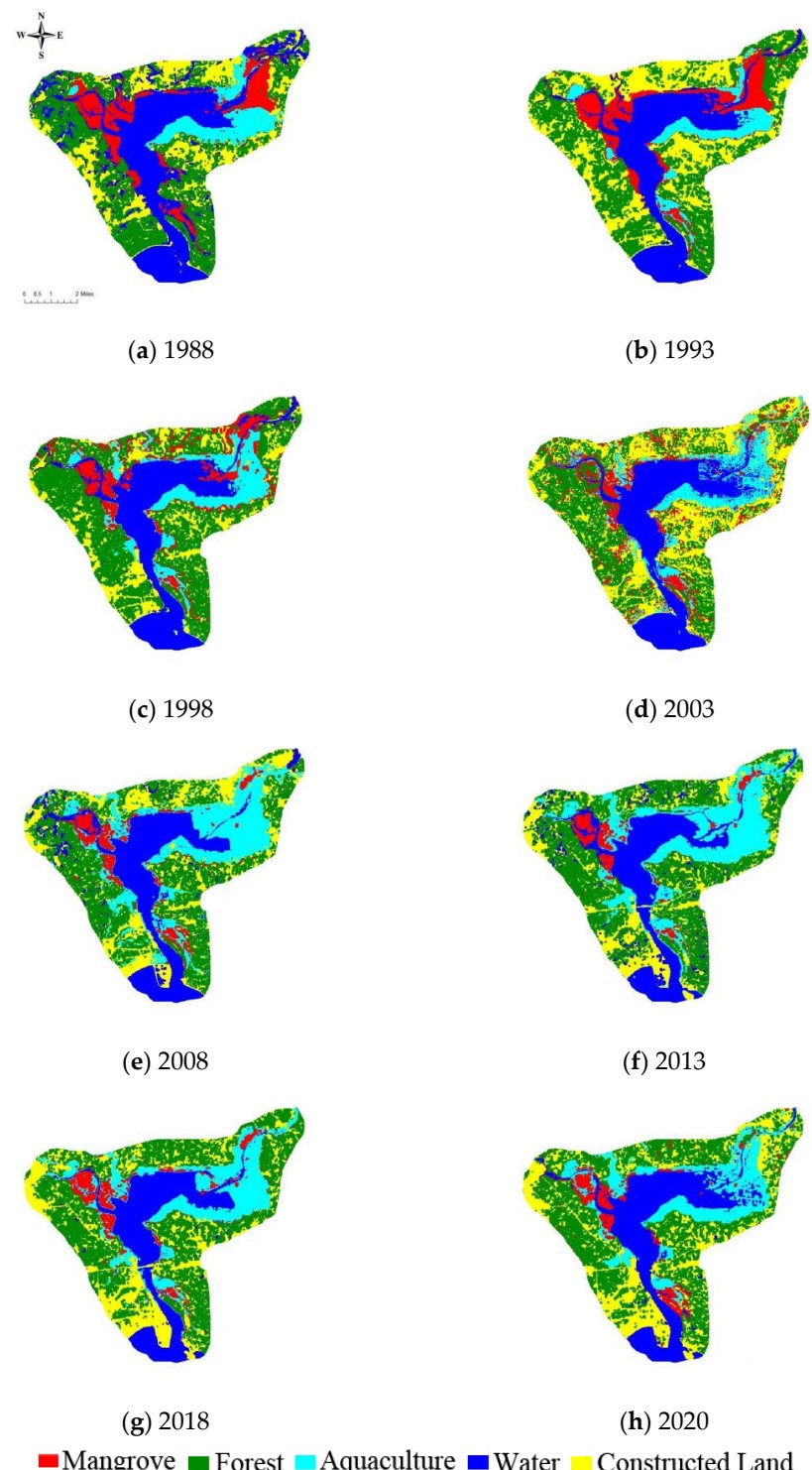

**Figure 2.** Spatial distribution of land use in the coastal zone of Qinglan Bay from 1988 to 2020.

**Table 4.** Dynamic changes in mangrove area of the Qinglan Bay from 1988 to 2020 (ha).

| Land Use | 1988 | 1993 | 1998 | 2003 | 2008 | 2013 | 2018 | 2020 |
|---|---|---|---|---|---|---|---|---|
| Mangrove | 1559.34 | 1619.28 | 1466.19 | 1019.61 | 805.5 | 602.01 | 849.33 | 737.37 |
| Forest | 4428.36 | 4461.03 | 5379.12 | 3533.67 | 4585.86 | 5049.18 | 5097.33 | 4804.65 |
| Aquaculture | 1153.35 | 1020.15 | 1747.08 | 1982.52 | 2926.98 | 3031.65 | 2474.46 | 2214.72 |
| Water | 4306.86 | 3286.35 | 3249.63 | 2928.15 | 3044.97 | 2962.53 | 2683.08 | 3127.32 |
| Constructed Land | 2377.53 | 3438.18 | 1983.15 | 4361.31 | 2461.95 | 2179.98 | 2721.15 | 2940.57 |

### 3.3. Estimation of Mangrove Carbon Stocks in Qinglan Bay Based on InVEST Model

The InVEST model carbon module saves the clustered data in Geotif format for analysis on the ArcGIS 10.2 platform. During this step, the processed land use data does not have an attribute table, therefore we used Data Management Tools in the toolbox and selected Build Raster Attribute Table to create a raster attribute table and to input land class names and code data to assign values to the raster map. Then, we opened the Carbon module in InVEST, inputting present and predicted data of land use and carbon pool in this step to calculate carbon stock.

### 3.4. Carbon Stock Distribution and Spatial Changes

In this study, the raster data of eight periods of land use types were extracted, and the net change of carbon storage over time can be calculated by the carbon module of the InVEST model, with each time period to be compared with each other. The land use type raster data and the carbon pool data were set as inputs into the InVEST model, and the spatial distribution of carbon storage in the coastal zone area of Qinglan Port, Hainan Province during 1988–2020 was acquired, as shown in Figure 3. By comparison, the differences in the spatial distribution of carbon stocks between two adjacent time phases is shown in Figure 4. As a result, the changes of the total carbon stock in eight-time phases are derived, as shown in Figure 5.

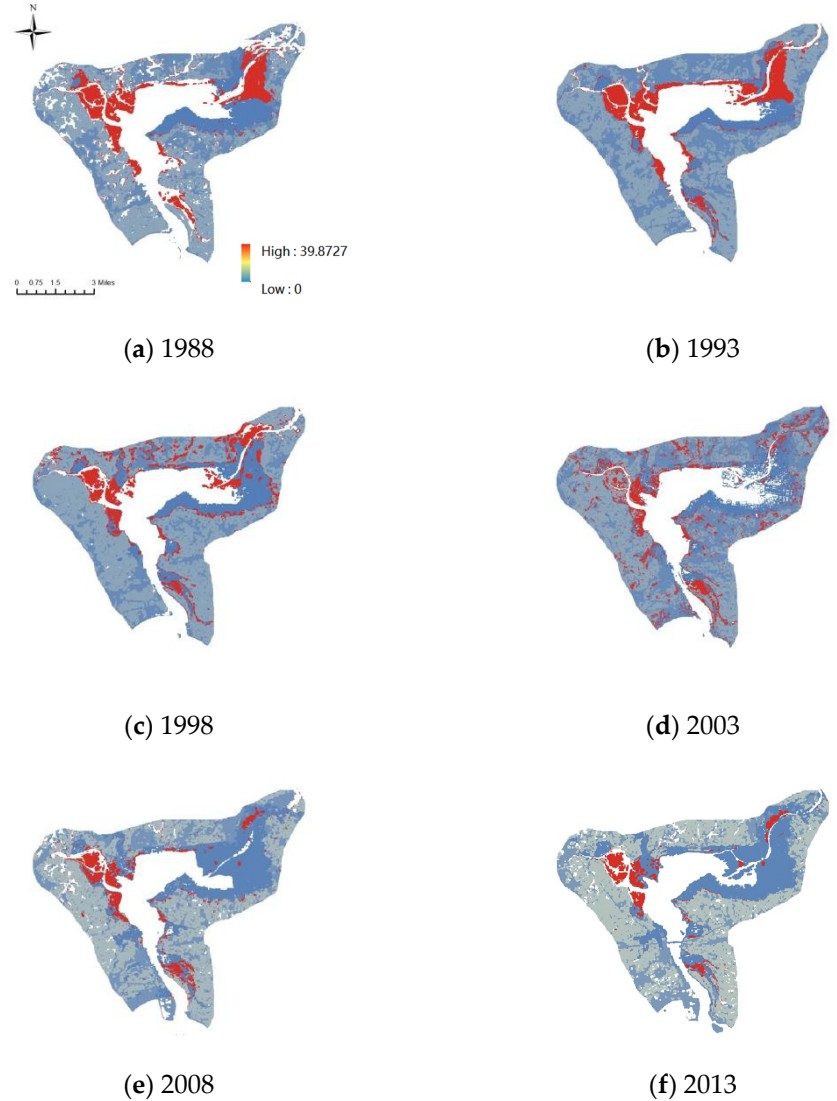

(**a**) 1988

(**b**) 1993

(**c**) 1998

(**d**) 2003

(**e**) 2008

(**f**) 2013

**Figure 3.** *Cont*.

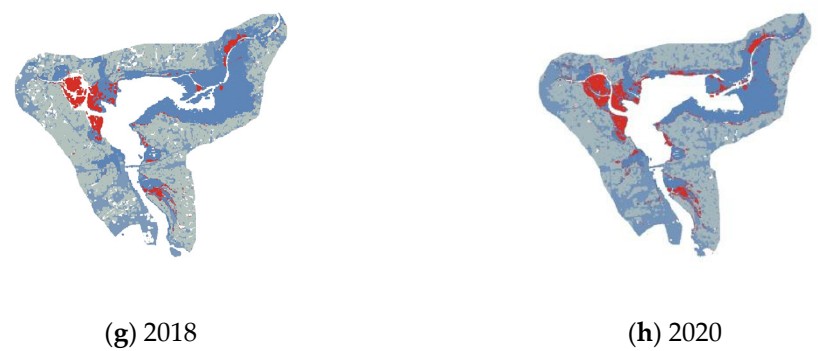

(**g**) 2018          (**h**) 2020

**Figure 3.** Spatial distribution of carbon stock in the coastal zone in Hainan Province from 1988 to 2020.

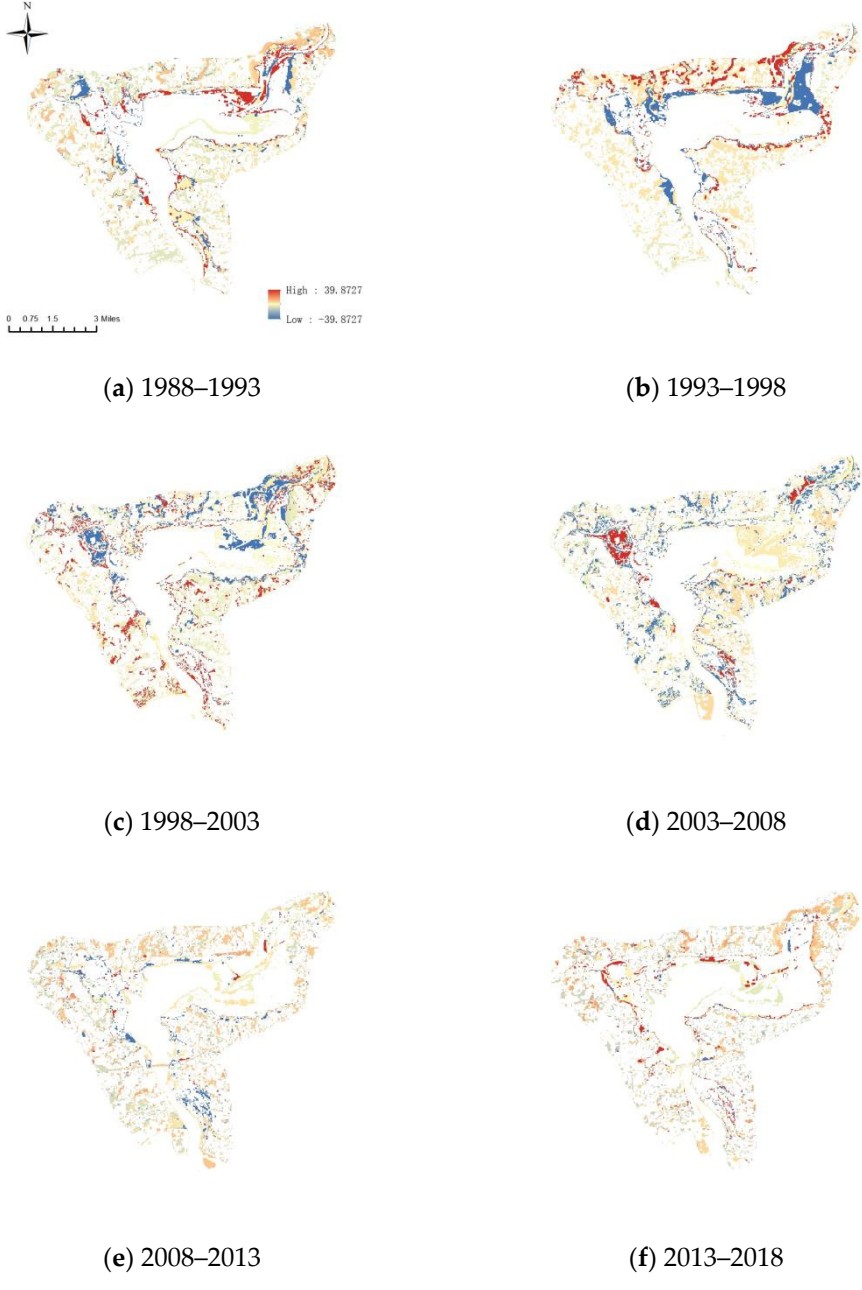

(**a**) 1988–1993        (**b**) 1993–1998

(**c**) 1998–2003        (**d**) 2003–2008

(**e**) 2008–2013        (**f**) 2013–2018

**Figure 4.** *Cont*.

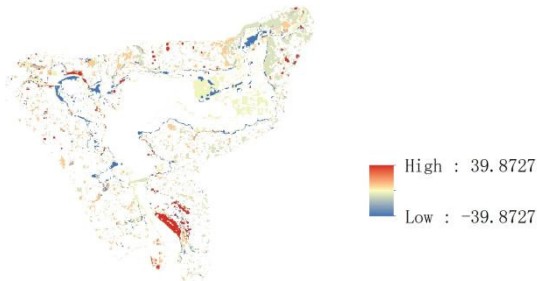

**(g)** 2018–2020

**Figure 4.** Spatial distribution map of two adjacent interphase carbon stock changes in the coastal zone of Qinglan Bay in Hainan Province from 1988 to 2020.

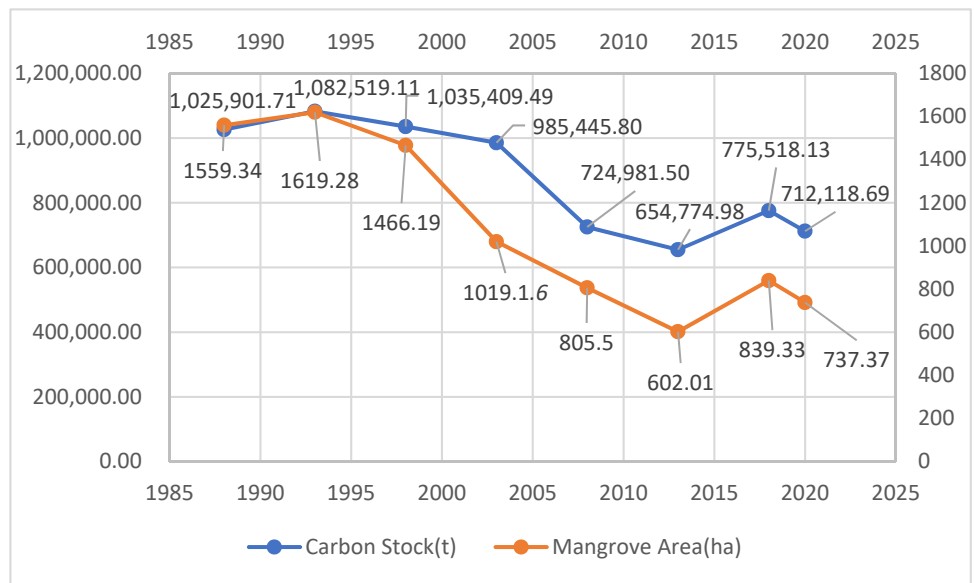

**Figure 5.** Total carbon stock changes with the mangrove cover change in Qinglan Port from 1988 to 2020 (ton).

The results compared by spatial analysis revealed that the curve of carbon stock change accordingly aligned with the mangrove area change. The highest distribution areas of carbon stocks were all located with the mangrove cover among the five land use types, and the mangrove vegetation distribution areas contributed the most to the carbon stocks. Through combining the spatial distribution maps of carbon stocks and land use between two adjacent time phases, the spatial and temporal evolution of carbon accumulation and carbon loss were analyzed. The results showed that carbon accumulation mainly occurred in the areas where the mangrove area increased, and carbon loss mainly occurred in the areas where the mangrove area decreased. In terms of time, the total carbon stock of the five land use types in the study area during 1988–2020 showed an overall decreasing trend from 1,025,901.71 tons to 712,118.69 tons annually, except for brief increases during 1988–1993 and 2013–2018, and the total carbon stock reached a maximum value of 1,082,519.11 tons in 1993 and a minimum of 712,118.69 tons in 2020. Since 2003, the ecological restoration project has slowed the decline of the mangrove area and the rate of carbon stock reduction in the Qinglan Bay mangrove wetland, thereby contributing to the stabilization and increase of this blue carbon sink.

## 4. Discussion

The InVEST model used in this study simplified the carbon cycle estimation and allowed it to be completed with relatively limited information. Nonetheless, the model also has its drawbacks in analysis. First, InVEST assumes that carbon stocks vary linearly over a given time period, or it may underestimate the carbon sink rates. Second, carbon storage estimates are influenced by land use types and carbon pool data. The more detailed the classification of land use types, the more concrete corresponding carbon pool data used and the closer the carbon storage to the actual values that would be estimated. In this study, only 8 periods of RS imagery in the past 30 years were selected. Therefore, in future studies, the accuracy of the research results can be further improved by adding more time RS imagery; increasing the land use types by function and planning; and classifying and analyzing the mangrove communities based on different plant categories and growth periods. Meanwhile, we propose that a more carbon emission monitoring system for the mangrove wetlands of Qinglan Bay can be developed in the near future by establishing RS dynamic stations for monitoring purposes in the mangrove area and adopting professional approaches.

## 5. Conclusions

In this study, the 3S technique was combined with the InVEST model to demonstrate the changes of blue carbon stock by investigating the changes of mangrove cover and land use and mangrove ecological restoration in Qinglan Bay mangrove forest in the past 30 years.

The results show that:

The transformation of carbon stock rate has aligned with the cover change rate of the mangrove forest in Qinglan Bay. The overall area of mangrove forests in the Qinglan Bay mangrove wetland in Hainan Province showed a decreasing trend during 1988–2020. The ecological restoration project slowed the decreasing rate of the mangrove forest area. The area of mangrove cover decreased from 1559.34 ha to 737.37 ha in its initial phase of which 52.71% was transformed into aquiculture, construction, and other land types. Most of the mangrove lands were turned into aquaculture water, and with the expansion of aquaculture water, the mangrove landscape gradually became fragmented, and some nearshore mangrove forests were transformed into constructed land.

The overall trend of the mangrove coastal carbon stock in Qinglan Bay from 1988 to 2020 is decreasing annually, from 1,025,901.71 tons to 712,118.69 tons. Among the five types of land use, mangroves have the largest contribution to carbon stock, and the highest value of carbon stock is distributed in the mangrove land type. Carbon accumulation mainly occurs in areas of increased mangroves, and carbon loss mainly occurs in areas of decreased mangroves. Since 2003, the ecological restoration project has slowed the decline of mangrove areas, which also slowed the rate of carbon stock reduction in the Qinglan Port mangrove wetland, having a positive impact on the stabilization and increase of blue carbon sink.

The results of this study provide a technical method to evaluate the carbon sink contribution of mangrove wetland ecological restoration in Qinglan Bay, Hainan Province. This method is a scientific and methodological innovation that can be used to monitor the carbon stork of mangrove cover changes in China on a large regional scale. In addition, this method provides a theoretical basis to select criteria for restoring and rebuilding mangrove habitats, and functions as an academic guidance for the development of scientific and systematic management strategies for mangrove ecological restoration on Hainan Island.

**Author Contributions:** Conceptualization, methodology, and writing—editing, P.J. and Z.Z.; writing—original draft and software, Z.Z., W.H., J.C. and Y.X.; funding acquisition, supervision, and project administration, P.J. All authors have read and agreed to the published version of the manuscript.

**Funding:** This research was funded by the National Social Science Found of China, grant number 21XGL019 and the Hainan Provincial Natural Science Foundation of China, grant number 721RC1048.

**Data Availability Statement:** The data presented in this study are openly available in Geospatial Data Cloud of the Computer Network Information Center, Chinese Academy of Sciences (http://www.gscloud.cn, 1 April 2020).

**Acknowledgments:** The authors gratefully acknowledge the two anonymous reviewers for their valuable comments and suggestions, which strengthened the quality of the paper substantially.

**Conflicts of Interest:** The authors declare no conflict of interest.

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
