# Peer review of "The Carbon Sink of Mangrove Ecological Restoration between 1988–2020 in Qinglan Bay, Hainan Island, China"

_forests, doi:10.3390/f13101547_

Round 1

Reviewer 1 Report

This a very interesting modeling for me, I really enjoy it. The authors did a great job in utilizing satellite images and gathering information to come up with a useful tool and model which can be utilized elsewhere with the same condition for a better mangrove ecosystem.

 I have very view comments that meant to upgrade the quality of work in order have a better published article.

- The authors rely on climate change for mangrove declining and carbon stock decrease with neglecting other environmental stress and anthropogenic activities, that may have some effects.

- The introduction was too long, it is better to reduce it.

- The result needs to be reconstructed. The authors should elaborate more on all figures and translate these maps into words. In figure 5 for example, a very interesting pattern was presented. In 1993 high density of mangroves with low carbon stock!!!! This should be explained. In 1998 overlapping and change positions, it should be clarified and so on.

- I will focus my last comment on VEST model which is an excellent tool. However, this model can estimate two things of carbon stock at the same time: storage and sequestration. What is the case in this paper?

To calculate storage, you need to sum; above and below ground biomass, soil, and dead wood. Are these components considered in this study?

Did you take all possible land use data in your consideration when building land use data for your model?

Hope these comments will trigger the author to improve their manuscript more which I consider well done paper.

Thank you all

Author Response

Dear Reviewer,

Thank you so much for your constructive suggestions and valuable comments on the manuscript with ID forests-1890848. We accepted these reasonable suggestions and revised our manuscript according to them. Meanwhile, we have revised and improved our results and conclusions section with regard to the comments and presenting the detailed revisions as follows.

  1. The authors rely on climate change for mangrove declining and carbon stock decrease with neglecting other environmental stress and anthropogenic activities, that may have some effects.

Reply: Thanks for your question. We have chosen climate change as our research focus mainly due to the carbon stock decrease. All mangrove declining and carbon stock decrease is come from anthropogenic activities, such as land transformation from mangrove land into construction, aquiculture, beach polder farming and land reclamation etc. as our analysis based on matrix transformation. That means the anthropogenic activities is one of the focal reasons which led to mangrove decline and carbon stock decrease. We have confirmed this situation in line 212-214.

  1. The introduction was too long, it is better to reduce it.

Reply: We have polished the introduction section to be shorter and more concise.

  1. The result needs to be reconstructed. The authors should elaborate more on all figures and translate these maps into words. In figure 5 for example, a very interesting pattern was presented. In 1993 high density of mangroves with low carbon stock!!!! This should be explained. In 1998 overlapping and change positions, it should be clarified and so on.

Reply: Thanks for your kind reminder. We have recalculated the data and redrew the figure and find out there some unit mistakes happened in original version and lead to different effect in 1993. We reduced the interruption water element and checked the data calculating, then redrew the new diagram already. In new diagram, the carbon stock curve is fitting to the mangrove area change. Please see figure 4 and figure 5. Meanwhile, we have reconstructed and have rewrote the result section already.

  1. I will focus my last comment on VEST model which is an excellent tool. However, this model can estimate two things of carbon stock at the same time: storage and sequestration. What is the case in this paper?

Reply: Thank you for your good suggestion! In this research, we focus on the spatial ecology, so we only estimate the carbon stock. In the future research, we can try to work on the carbon sequestration and get new achievement.

  1. To calculate storage, you need to sum; above and below ground biomass, soil, and dead wood. Are these components considered in this study?

Reply: Yes, In this study, the carbon storage we had calculated includes the above, below ground biomass, soil and dead wood. Please see line 160-164.  

  1. Did you take all possible land use data in your consideration when building land use data for your model?

Reply: The carbon storage calculation focused on the specific area where mangrove was growing. So, we did take all possible land use data in our consideration including constructed land in our study area because there is some mangrove land transformed into constructed land. Please see line 212-214.

  1. Hope these comments will trigger the author to improve their manuscript more which I consider well done paper.

Reply: Very appreciate for your valuable advice which gives us much more inspiration. Thank you very much.

At last, we rewrote and improved our results and conclusions already according to all above revisions.

We gratefully acknowledge you for your valuable comments and suggestions, which strengthened the quality of the paper substantially.

Best regards,

Prof., Dr. Peihong Jia (贾培宏)

Department of Land Resource Management

School of Public Administration

Hainan University

58 Renmin Avenue, Haikou

Hainan Province, 570228

Email: jiaph@hainanu.edu.cn

https://hd.hainanu.edu.cn/zhengguan/info/1073/5681.htm

Reviewer 2 Report

The investigation is carefully designed, implemented, and reported. The literature is discussed in a comprehensive manner.

The Authors should carefully review and revise all numbers, units, and multipliers. There is a lot of confusion this reviewer is unable to sort out.

For example, on line 63 there is a quantity “1023 Mg Chm-2”. Does this possibly mean “1023 Mg [C] (hm)-2”?
It is worth noting that   hm-2 = 106 (hm)-2.

Then, in table 2 there are carbon storage numbers, approximately half of the number given on line 63. Are they possibly in the same units?

Author Response

Dear Reviewer,

Thank you so much for your constructive suggestions and valuable comments on the manuscript with ID forests-1890848. We accepted these reasonable suggestions and revised our manuscript according to them. Meanwhile, we have revised and improved our results and conclusions section with regard to the comments and presenting the detailed revisions as follows.

  1. The investigation is carefully designed, implemented, and reported. The literature is discussed in a comprehensive manner. The Authors should carefully review and revise all numbers, units, and multipliers. There is a lot of confusion this reviewer is unable to sort out.

Reply: Thanks, we appreciate your scientific attitude. We did find some numbers and units mistakes after carefully checking all through the manuscript. We are taking it seriously and had corrected them and the related diagrams accordingly. Please check table2, figure 4 and figure 5.

  1. For example, on line 63 there is a quantity “1023 Mg Chm-2”. Does this possibly mean “1023 Mg [C] (hm)-2”?
    It is worth noting that   hm-2= 106 (hm)-2.

Reply: Thank you so much for your careful review. We have checked all the number and units cautiously already and have revised them as the international criteria. Please check table2, line 340-344 etc. 

  1. Then, in table 2 there are carbon storage numbers, approximately half of the number given on line 63. Are they possibly in the same units?

Reply: Thanks for your reminder. There are some unit mistakes existed in table2. We have confirmed and revised them.

  1. Lastly, we rewrote and improved our results and conclusions as well according to all the revisions before.

Reply: Thank you again for your valuable suggestions. We rewrote and improved our results and conclusions already according to all above revisions.

We gratefully acknowledge you for your valuable comments and suggestions, which strengthened the quality of the paper substantially.

Best regards,

Prof., Dr. Peihong Jia (贾培宏)

Department of Land Resource Management

School of Public Administration

Hainan University

58 Renmin Avenue, Haikou

Hainan Province, 570228

Email: jiaph@hainanu.edu.cn

https://hd.hainanu.edu.cn/zhengguan/info/1073/5681.htm 

Round 2

Reviewer 1 Report

The authors did a great job of correcting and revising the manuscript according to the given comments. Therefore, I endorse the manuscript for publication.